# Communication-efficient federated learning via knowledge distillation

Chuhan Wu [1], Fangzhao Wu [2✉], Lingjuan Lyu [3], Yongfeng Huang [1✉] & Xing Xie [2]

Federated learning is a privacy-preserving machine learning technique to train intelligent models from decentralized data, which enables exploiting private data by communicating local model updates in each iteration of model learning rather than the raw data. However, model updates can be extremely large if they contain numerous parameters, and many rounds of communication are needed for model training. The huge communication cost in federated learning leads to heavy overheads on clients and high environmental burdens. Here, we present a federated learning method named FedKD that is both communication-efficient and effective, based on adaptive mutual knowledge distillation and dynamic gradient compression techniques. FedKD is validated on three different scenarios that need privacy protection, showing that it maximally can reduce 94.89% of communication cost and achieve competitive results with centralized model learning. FedKD provides a potential to efficiently deploy privacy-preserving intelligent systems in many scenarios, such as intelligent healthcare and personalization.

[1] Department of Electronic Engineering, Tsinghua University, Beijing 100084, China. [2] Microsoft Research Asia, Beijing 100080, China. [3] Sony AI, 1-7-1 Konan Minato-ku, Tokyo 108-0075, Japan. ✉email: fangzwu@microsoft.com; yfhuang@tsinghua.edu.cn

The boom of deep learning in recent years has greatly benefited humans in various fields such as intelligent healthcare[1] and personalized web applications[2]. In most scenarios, abundant and accessible data is usually a prerequisite for training high-quality deep learning models[3]. Thus, many deep learning-based approaches rely on centralized data storage for centralized model training[4]. However, many types of data are privacy-sensitive in nature, such as medical data kept by hospitals and user behavior data stored on users' personal devices[5,6]. With the increasing attention on privacy protection, many data protection regulations like GDPR have strictly restricted sharing privacy-sensitive data among different clients or platforms[7]. Thus, it can be difficult to collect and aggregate decentralized data into a unified dataset for centralized model training, which poses great challenges to the applications of many existing deep learning techniques without privacy awareness[8].

Federated learning is a recent technique to learn intelligent models from decentralized private data[4], which has been widely used in various applications that involve privacy-sensitive data such as intelligent healthcare[9,10], personalized recommendation[11], and mobile keyboard prediction[12]. Federated learning offers a privacy-aware learning paradigm that does not require raw data sharing, but allows participants to collaboratively construct a better global model by only sharing their local model updates[4]. Instead of collecting and storing data in a centralized database, in federated learning private data is locally stored on different clients[13]. Each client keeps a local model and computes the model updates by learning on its local data. In each iteration, a number of clients first compute their local model updates and upload them to a server, which aggregates the local model updates into a global one to update its maintained global model. Then, the server distributes the global update to each client to conduct a local model update. This process is iteratively executed for many rounds until the model converges. Since the communicated model updates usually contain much less private information than the raw data, user privacy can be protected to some extent when training machine learning models on decentralized data[13].

In federated learning, the server and clients need to intensively communicate the model updates during model training[14]. Thus, the communication cost is enormous if the model is in large size. Unfortunately, the state-of-the-art deep learning models (e.g., pretrained language models) have become more and more giant in recent years, which contain billions and even trillions of learnable parameters[15,16]. It can lead to a huge overhead when communicating these cumbersome models, which is especially non-practical for decentralized clients with relatively limited communication bandwidth and throughput[17]. Thus, though quite promising in terms of their performance, big models are rarely deployed in conventional federated learning systems[18].

In recent years, researchers have been making great efforts in improving the communication efficiency of federated learning[19]. One potential way is gradient compression[17,20,21], which can directly reduce the size of model updates. However, they usually suffer from a heavy performance sacrifice when the compression ratio is required to be very large[22]. In addition, compressing the global model updates may also degrade the model capability in handling the heterogeneity of decentralized data due to the limited model capacity[23]. Another widely used paradigm in communication-efficient federated learning is codistillation[23–28]. Instead of transferring model updates, it only communicates the local model prediction on a public dataset that is shared among different clients, which can reduce the communication cost if the size of local model is larger than the public data. Unfortunately, in many real-world scenarios, such as personalized recommendation and electronic medical records understanding, the data are highly privacy-sensitive and may not be able to share nor exchange (even

after anonymization). Thus, both effective and communication-efficient federated learning without the aid of additional data is an important but still unresolved challenge.

In this work, we present a communication-efficient federated learning method based on knowledge distillation, named FedKD. Our method is mainly focused on cross-silo federated learning where the clients have relatively richer computing resources and larger local data volume than personal devices. Instead of directly communicating the large models between the clients and server, in FedKD there are a small model (mentee) and a large model (mentor) that learn and distill knowledge from each other, where only the mentee model is shared by different clients and learned collaboratively, which can effectively reduce the communication cost. In this way, different clients have different local mentor models, which can better adapt to the characteristics of local datasets to achieve personalized model learning. Motivated by the Protégé Effect[29] ("while we teach, we learn", Seneca the Younger) in human learning, we propose an adaptive mutual distillation method to enable the local mentor and mentee models to reciprocally learn from the knowledge distilled from their predicted soft labels and intermediate results, where the distillation intensity is self-adaptively controlled by the correctness of their predictions. In addition, to further reduce the communication cost when exchanging the mentee model updates, we introduce a dynamic gradient approximation method based on singular value decomposition (SVD) to compress the communicated gradients with dynamic precision, which can achieve a promising tradeoff between communication cost and model accuracy. We evaluate FedKD on five benchmark datasets for three different real-world applications that require privacy protection, including personalized news recommendation, adverse drug reaction (ADR) mentioning text detection, and medical named entity recognition (NER). Experimental results demonstrate that FedKD can maximally reduce 94.89% of the communication cost of standard federated learning methods, which has the potential to support federated big model learning in practice. In addition, FedKD achieves comparable or even better performance than centralized model learning, and outperforms many other communication-efficient federated learning methods. FedKD also shows great abilities in handling non-independent identically distributed (non-IID) data on different clients due to the personalization effect of local mentor models, which may provide a potential direction to heterogeneous data mining.

## Results

**Performance evaluation.** Our experiments are conducted in three tasks that involve user data. The first task is personalized news recommendation, which needs to predict whether a user will click a candidate news based on the user interest inferred from historical news click behaviors. In this task we use the MIND[30] dataset. The second one is ADR mentioning text detection, which is a binary classification task. We use the dataset released by the 3rd shared task of the SMM4H 2018 workshop[31]. We denote this dataset as ADR. On these two datasets, to simulate the scenario where private data is decentralized among different clients, we randomly divide the training data into four folds and assume that each fold is locally stored by a different client. We use the average performance of local mentor models on the local data as the overall performance. The third task is medical NER, which aims to extract medical entities from plain texts and classify their types. We conduct experiment on three public benchmark medical NER datasets, e.g., CADEC[32], ADE[33] and SMM4H[34]. These three datasets have very different characteristics such as data collection sources and label tagging schemes. Thus, we regard that each of them is kept by a client to evaluate the performance of different federated learning methods in handling non-IID data. More details of the datasets

and metrics used for evaluation are provided in the Datasets section in Supplementary Information.

In our experiments, on each client we use the Base version of UniLM[35] as the local mentor model. We use its submodels with the first four or two transformer layers as the mentee models. We compare FedKD with several groups of baselines. The first baseline group includes (1) UniLM (Local), (2) UniLM (Cen), and (3) UniLM (Fed), which stand for learning the full UniLM model on local data only, on centralized data, or on decentralized data with the FedAvg[4] framework, respectively. Note that we do not use the classic FedSGD[17] method because its communication cost is unacceptable, i.e., tens of trillionbytes per client. The second group finetunes different compressed models with federated learning using the FedAvg algorithm, including: (4) DistilBERT[36], (5) BERT-PKD[37], (6) TinyBERT[38], (7) MiniLM[39], and (8) UniLM$_{4/2}$ (the first four or two layers of UniLM). For DistilBERT, BERT-PKD, TinyBERT, and MiniLM, we compare their officially released six-layer and four-layer models. The third group contains several gradient compression methods for federated learning, including: (9) FetchSGD[21], (10) FedDropout[20], (11) SCAFFOLD[40], and (12) FedPAQ[41]. We compare FedPAQ with 16-bit or 8-bit precision levels. These four baselines are representative ones that use different ways to improve communication efficiency, including sketched update, reducing parameters, convergence acceleration, and parameter quantization. The metrics used for evaluation are AUC, MRR, nDCG@5 and nDCG@10 on MIND, and are precision, recall and Fscore of the positive class on ADR (explanations are given in the Experimental Settings section of Supplementary Information). We report the model performance and communication cost on the MIND and ADR datasets (Table 1). From the results, we find the performance of UniLM (Local) is inferior to other methods that can exploit decentralized data, which is mainly due to the reason that local data on a single client may be insufficient to learn a strong model. Although UniLM (Fed) achieves comparable results with centralized learning, its communication cost for model learning is huge (e.g., over 2GB for each client on MIND), which may hinder its applications in real-world systems. Both model compression methods and gradient compression methods can reduce the communication cost to certain extents, but they either have major performance decreases or can only save very limited communication resources. Different from them, the performance of FedKD is even comparable with learning a big model on centralized data. For example, the four-layer FedKD can achieve a 71.0% AUC on MIND and 60.7% Fscore on ADR, while the corresponding scores of finetuning the original big model on centralized data are 71.0% and 60.8%. Further two-sided t-test also shows that their differences are not significant $p > 0.1$. This is because in FedKD there are multiple mentor models on different decentralized clients for personalized learning and knowledge distillation. In addition, FedKD can save up to 94.63% and 94.89% of communication cost on MIND and ADR, respectively, which is more communication-efficient than other compared federated learning-based methods. This is because FedKD can learn useful knowledge from the sophisticated local mentor models to improve the model performance, and can reduce the communication cost by exchanging the updates of a small mentee model and meanwhile compress the gradients using SVD with a dynamic precision. These results show that FedKD can effectively reduce the communication cost of federated learning while keeping promising model performance.

We also evaluate the performance of different federated learning methods in the medical NER task to compare their effectiveness in handling heterogeneous data (Fig. 1). We find FedKD outperforms other compared methods with a substantial margin. For example, the absolute improvement on the SMM4H dataset over the best-performed baseline is 3.9%. Different from many other baselines that use globally shared models on all clients, the local mentor models in FedKD on different clients are personalized. These local mentor models can better adapt to the heterogeneous characteristics of local data on different clients. Thus, FedKD has a greater potential than many non-personalized

**Table 1 Performance (with standard deviations) and communication cost per client of different methods on MIND and ADR.**

| Methods | MIND | | | | | ADR | | | |
|---|---|---|---|---|---|---|---|---|---|
| | AUC | MRR | nDCG@5 | nDCG@10 | Comm. cost per client | Precision | Recall | Fscore | Comm. cost per client |
| UniLM (Local) | 68.8 ± 0.5 | 33.5 ± 0.4 | 36.6 ± 0.5 | 42.4 ± 0.6 | – | 53.2 ± 1.3 | 54.6 ± 1.4 | 53.9 ± 1.1 | – |
| UniLM (Cen) | 71.0 ± 0.1 | 35.8 ± 0.1 | 39.0 ± 0.1 | 44.8 ± 0.1 | – | 60.3 ± 0.7 | 61.6 ± 0.8 | 60.8 ± 0.4 | – |
| UniLM (Fed) | 70.9 ± 0.3 | 35.7 ± 0.2 | 38.9 ± 0.3 | 44.7 ± 0.4 | 2.05GB (1.0×) | 59.1 ± 0.6 | 62.3 ± 0.6 | 60.6 ± 0.4 | 1.37GB (1.0×) |
| DistilBERT$_6$ | 69.3 ± 0.2 | 34.0 ± 0.2 | 37.5 ± 0.2 | 43.0 ± 0.1 | 1.03GB (2.0×) | 56.8 ± 0.8 | 59.2 ± 0.8 | 57.9 ± 0.5 | 0.69GB (2.0×) |
| DistilBERT$_4$ | 69.0 ± 0.2 | 33.7 ± 0.1 | 37.0 ± 0.1 | 42.6 ± 0.2 | 0.69GB (3.0×) | 56.5 ± 0.9 | 58.4 ± 1.1 | 57.1 ± 0.7 | 0.46GB (3.0×) |
| BERT-PKD$_6$ | 69.6 ± 0.2 | 34.4 ± 0.3 | 37.7 ± 0.3 | 43.4 ± 0.2 | 1.03GB (2.0×) | 56.9 ± 0.9 | 60.4 ± 0.8 | 58.4 ± 0.6 | 0.69GB (2.0×) |
| BERT-PKD$_4$ | 69.2 ± 0.2 | 33.8 ± 0.2 | 37.1 ± 0.3 | 42.9 ± 0.3 | 0.69GB (3.0×) | 56.3 ± 1.1 | 59.9 ± 0.7 | 58.0 ± 0.6 | 0.46GB (3.0×) |
| TinyBERT$_6$ | 69.7 ± 0.2 | 34.5 ± 0.2 | 37.9 ± 0.1 | 43.5 ± 0.2 | 1.03GB (2.0×) | 57.4 ± 0.8 | 60.5 ± 0.6 | 58.6 ± 0.5 | 0.69GB (2.0×) |
| TinyBERT$_4$ | 69.4 ± 0.3 | 33.9 ± 0.3 | 37.5 ± 0.2 | 43.1 ± 0.2 | 0.17GB (12.1x) | 57.0 ± 0.7 | 59.9 ± 1.2 | 58.3 ± 0.7 | 0.12GB (11.4x) |
| MiniLM$_6$ | 70.0 ± 0.1 | 34.9 ± 0.1 | 38.1 ± 0.1 | 43.8 ± 0.2 | 1.03GB (2.0×) | 55.9 ± 0.9 | 62.1 ± 0.8 | 58.8 ± 0.6 | 0.69GB (2.0×) |
| MiniLM$_4$ | 69.6 ± 0.2 | 34.0 ± 0.2 | 37.6 ± 0.2 | 43.2 ± 0.3 | 0.17GB (12.1x) | 56.8 ± 0.9 | 60.5 ± 1.0 | 58.6 ± 0.6 | 0.12GB (11.4x) |
| UniLM$_4$ | 69.6 ± 0.1 | 34.4 ± 0.2 | 37.7 ± 0.1 | 43.4 ± 0.2 | 0.69GB (3.0×) | 56.1 ± 0.9 | 60.6 ± 0.9 | 58.2 ± 0.5 | 0.46GB (3.0×) |
| UniLM$_2$ | 68.9 ± 0.2 | 33.6 ± 0.2 | 36.8 ± 0.2 | 42.5 ± 0.1 | 0.35GB (5.9×) | 53.8 ± 0.8 | 59.1 ± 1.0 | 56.3 ± 0.6 | 0.24GB (5.7×) |
| FetchSGD | 70.5 ± 0.4 | 35.2 ± 0.3 | 38.2 ± 0.3 | 44.0 ± 0.4 | 0.51GB (4.0×) | 57.5 ± 0.9 | 60.4 ± 1.1 | 59.0 ± 0.8 | 0.34GB (4.0×) |
| FedDropout | 70.5 ± 0.2 | 35.1 ± 0.2 | 38.3 ± 0.3 | 44.2 ± 0.3 | 1.23GB (1.7×) | 57.8 ± 1.0 | 61.0 ± 0.8 | 59.4 ± 0.6 | 0.82GB (1.7×) |
| SCAFFOLD | 70.7 ± 0.1 | 35.4 ± 0.2 | 38.7 ± 0.1 | 44.5 ± 0.2 | 2.73GB (0.8×) | <u>58.8</u> ± 0.8 | 61.9 ± 0.9 | 60.3 ± 0.5 | 2.74GB (0.5×) |
| FedPAQ (16-bit) | <u>70.8</u> ± 0.1 | <u>35.5</u> ± 0.1 | <u>38.8</u> ± 0.2 | <u>44.7</u> ± 0.2 | 1.03GB (2.0×) | 58.4 ± 1.1 | 61.2 ± 0.8 | 59.7 ± 0.7 | 0.69GB (2.0×) |
| FedPAQ (8-bit) | 70.2 ± 0.3 | 35.0 ± 0.3 | 38.1 ± 0.3 | 44.0 ± 0.4 | 0.51GB (4.0×) | 56.5 ± 1.2 | 59.4 ± 0.9 | 57.9 ± 0.8 | 0.34GB (4.0×) |
| FedKD$_4$ | **71.0** ± 0.1 | **35.6** ± 0.1 | **38.9** ± 0.1 | **44.8** ± 0.1 | 0.19GB (10.8×) | **59.4** ± 0.6 | **62.8** ± 0.9 | **60.7** ± 0.5 | 0.12GB (11.4x) |
| FedKD$_2$ | 70.5 ± 0.1 | 35.3 ± 0.2 | 38.6 ± 0.1 | 44.3 ± 0.2 | 0.11GB (**18.6×**) | 58.2 ± 0.7 | <u>62.4</u> ± 0.9 | <u>59.8</u> ± 0.6 | 0.07GB (**19.6×**) |

Local: learning model only on local data. Cen: learning on centralized datasets. Fed: a standard federated learning method FedAvg. Subscript numbers indicate the number of model hidden layers. The best Fscore (bold) of federated methods on ADR is significantly better than the second best one (underline) at the level of $p < 0.05$. The results show that the standard federated learning method can lead to heavy communication overheads, and FedKD can achieve promising results with much lower communication costs than FedAvg and other communication-efficient federated learning methods.

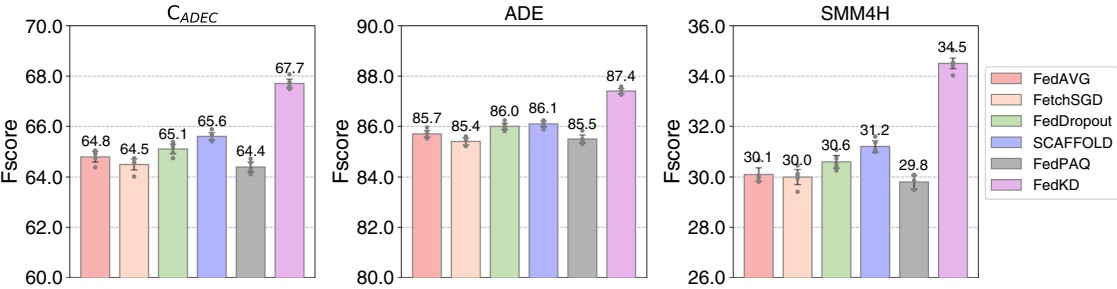

**Fig. 1 Performance comparison of different federated learning methods in the medical NER task.** The histogram height represents the Fscore of the corresponding method. The error bars represent the mean values with 95% confidence intervals ($n = 5$ independent experiments). The results show that FedKD is more effective than other compared federated learning methods in handling non-IID data.

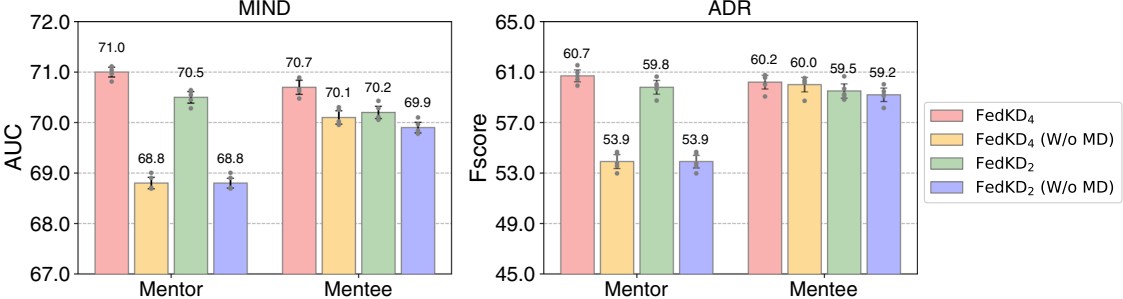

**Fig. 2 Influence of mutual distillation on the mentee and mentor models.** The mean values of AUC scores on MIND and F1 scores on ADR as well as their 95% confidence intervals are illustrated ($n = 5$ independent experiments). We compare the performance of mentors and the four-layer or two-layer mentee models when mutual distillation (MD) is used or not. The results show that mutual distillation can improve the performance of both mentee and mentor models, which is because useful knowledge can be reciprocally transferred between the mentee and mentor.

FL methods in handling data heterogeneity, i.e., training data on different clients is non-IID.

**Model effectiveness**. Next, we verify the effectiveness of our proposed adaptive mutual distillation method in FedKD. We first compare the performance of mentor and mentee models in FedKD trained with or without mutual distillation (the mentor model is only learned on local data, and the mentee still learns from local mentors) (Fig. 2). We observe that mutual distillation can effectively improve the performance of both mentor and mentee models with different sizes, especially the mentor model. This is because the mentor models have deeper architectures, and its encoded knowledge can benefit the mentee. Since the mentor models are learned on local data only, the useful knowledge encoded by the mentee can also bring complementary information to help mentors break the limitation of the amount of local labeled data. Since we observe that local mentors slightly outperform the mentee, we choose to use the mentor models for inference in the test stages.

We further compare FedKD and its variants by removing the adaptive mutual distillation loss, the adaptive hidden loss or the adaptive loss weighting method (Fig. 3). Note that we report the performance of mentor models. We can see that both adaptive mutual distillation and adaptive hidden losses are useful for improving the model performance. In addition, the performance is suboptimal when the adaptive loss weighting method is removed (this variant is similar to the standard mutual distillation[42]). This is because weighting the distillation and hidden losses can be aware of the correctness of model predictions, which may help distill higher-quality knowledge and meanwhile mitigate the risk of overfitting.

**Analysis of dynamic gradient approximation**. We then present some analysis of our proposed SVD-based gradient compression method with dynamic precision, which uses different singular value energy cutoff threshold $T$ at different model training stages. We show

the cumulative energy distributions of singular values of different parameter gradient matrices in the UniLM model (Fig. 4a, b). We find all kinds of parameter matrices in the model are low-rank, especially the parameters in the feed-forward network. This is because large deep learning models have the over-parameterization problem[43], and thereby the model parameters are low-rank. Thus, the communication cost can be greatly reduced by compressing the low-rank gradient matrices. In addition, we find a new phenomenon that the singular value energy is more concentrated at the beginning than the end of training. This may be because when the model is not well tuned, the gradients may have more low frequency components that aim to push the model to converge more quickly. However, when the model gets to converge, the updates of model parameters are usually subtle, which yields more high-frequency components. We also show the evolution of the number of required singular values under $T = 0.95$ (Fig. 4c). Note that the model is trained in a centralized way to get a nearly continuous curve because FedKD only has a small number of updates. We can see that more singular values need to be retained to achieve the same energy threshold. Thus, we choose to set a higher energy threshold after more iterations to learn accurate models for FedKD (discussions on the selection of energy threshold are shown in Supplementary Fig. 2).

## Discussion

In this work, we present a communication-efficient federated learning method named FedKD, which can effectively reduce the communication cost of federated learning without much performance sacrifice. The core idea of FedKD is exchanging the local updates of a small model rather than the original big models to reduce the communication cost. Since simply reducing the model size will lead to a notable performance decrease, we incorporate an adaptive mutual distillation method to encourage the small model (mentee) and the local big models (mentors) to reciprocally learn from each other. The local mentor models can adapt to the unique

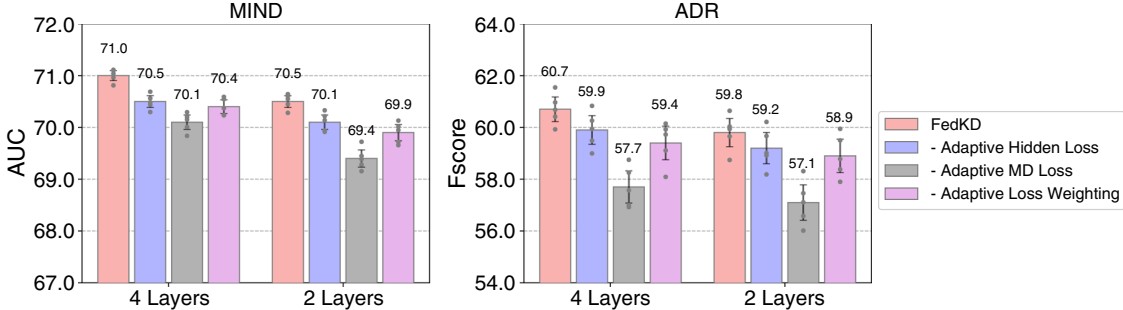

**Fig. 3 Effectiveness of the adaptive mutual distillation techniques in FedKD.** The mean results with 95% confidence intervals are presented ($n = 5$ independent experiments). We compare the results by removing the adaptive hidden loss, adaptive mutual distillation loss or the adaptive loss weighting method from FedKD. Adaptive hidden loss: the distillation loss function that aims to transfer knowledge encoded by the hidden states and intermediate results of models, where the loss intensity is weighted by the prediction loss of mentee and mentor models. Adaptive MD loss: the distillation loss function that aims to distill knowledge from the output soft labels of models, and its intensity is also controlled by the prediction loss. Adaptive loss weighting: the mechanism that weights the two distillation losses based on the summation of cross-entropy losses of the mentor and mentee models. We find the performance drops when either of them is removed, which verifies their contributions to federated model learning and distillation.

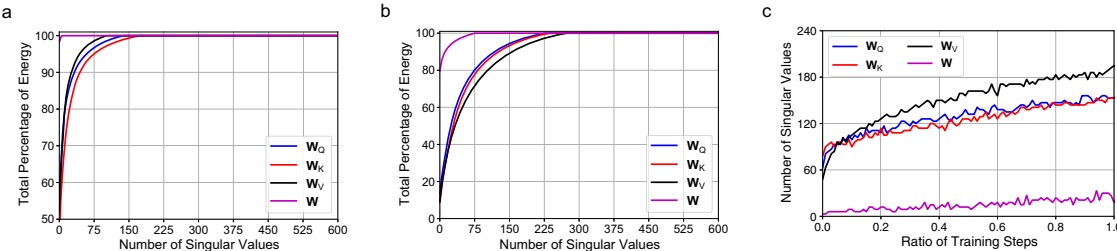

**Fig. 4 Analysis of gradient energy distribution. a** Cumulative energy distributions of singular values of different types of parameter gradient matrices at the beginning of model training. **b** Cumulative energy distributions of singular values at the end of model training. **c** Evolution of the number of required singular values during model training under a singular value energy cutoff threshold $T = 0.95$. $\mathbf{W}_Q$: query parameters, $\mathbf{W}_K$: key parameters, $\mathbf{W}_V$: value parameters, $\mathbf{W}$: feed-forward network parameters. The results show that the gradients are usually low-rank, and they have more high-frequency components after more rounds of model training. Thus, a relatively higher energy threshold needs to be used to keep higher gradient precision at the end of model training for better model accuracy.

characteristics of data on their clients to achieve personalization, and the mentee model is collaboratively learned on decentralized data to convey the knowledge distilled from multiple clients. To distill high-quality knowledge among mentors and the mentee, we further introduce an adaptive loss weighting mechanism based on their prediction correctness. Our adaptive mutual distillation approach is a general method that is not peculiar to the federated learning scenario. It can provide a new direction to compress large deep learning models and meanwhile keep high accuracy, which can serve as a fundamental technique to empower efficient deep learning. It also enables learning personalized models on non-IID data, which can alleviate the common challenge of data heterogeneity in real-world scenarios. In addition, since it is consistent with human's learning patterns, it may provide a new perspective to facilitate the understanding of knowledge transfer phenomena in artificial intelligence systems.

In deep learning models their updates are usually low-rank, and can be further compressed without hurting much precision. To further reduce the size of communicated model updates, we propose an SVD-based method to compress the model updates by exploiting their low-rank properties. However, using a fixed approximation precision may lead to a suboptimal performance because the model parameters may change very subtly when the model nearly converges. We demonstrate that the energy of singular values of model updates becomes less concentrated and there are more high-frequency components after more rounds of iterations. Based on this finding, we introduce a compression method with dynamic approximation precision to better adapt to this characteristic to learn more accurate models. The proposed method is quite simple and effective, and is compatible with

many other gradient compression methods. Our findings can not only inspire developing more effective model compression methods, but also can help researchers understand the inherent learning mechanism of deep learning models.

We conducted extensive experiments on five real-world datasets under three different scenarios, including personalized news recommendation, ADR mentioning text detection, and medical NER. The results show that FedKD can save up to 94.89% of communication cost, and can achieve comparable performance with centralized model learning (the minimal performance loss is less than 0.1%). In addition, the experimental results on the medical NER task with non-IID data show that FedKD outperforms many other federated learning methods with a substantial margin. It indicates that FedKD has a greater potential in overcoming the barrier between heterogeneous data through the personalization effect brought by different local mentor models.

The framework we proposed can be used as a benchmark for communication-efficient and privacy-preserving deep learning, which has the potential to be applied in various scenarios that involve private user data, such as personalization, intelligent healthcare and financial services. The proposed approach is especially suitable for being deployed in clients with limited communication resources, i.e., edge and personal devices, to save their communication resources like network bandwidth and throughput and meanwhile reduce the burden on the environment. It is beneficial for attracting more parties to participate in federated learning, which can empower the development of swarm intelligence.

In our future work, we will explore combining FedKD with other communication-efficient federated learning techniques to further reduce the communication cost. In addition, we plan to

study how to learn large models on low-resource devices to better support the application of FedKD in cross-device settings. We will also explore the application of FedKD in other real-world scenarios that need privacy-preserving deep learning. Furthermore, we plan to deploy FedKD in real-world personalization systems to learn intelligent user profiling models to serve users in a privacy-preserving and efficient way.

However, our work also has the following limitations. First, if the local data on each client is too scarce, it may be difficult to learn accurate mentor models. Thus, parameter-efficient deep learning techniques need to be combined with FedKD when local data are very limited. Second, in our approach we assume that the server is trusted and the communication channels are safe. Thus, if the server is hacked or there are malicious attackers, some private information encoded in model updates may not be fully protected. Third, FedKD will slightly increase the computational cost of local clients since they simultaneously maintain a mentor and an additional mentee model, which may not be suitable for edge and mobile devices. Thus, some computation accelerating methods such as low-bit quantization can be combined with FedKD to alleviate computational cost in practice.

## Methods

In this section, we introduce the details of our communication-efficient federated learning approach based on knowledge distillation (FedKD). We first present a definition of the problem studied in this paper, then introduce the details of our approach, and finally present some discussions on the computation and communication complexity of our approach.

**Problem definition.** In our approach, we assume that there are $N$ clients that locally store their private data, where the raw data never leaves the client where it is stored. We denote the dataset on the $i$th client as $D_i$. In our approach, each client keeps a large local mentor model $T_i$ with a parameter set $\Theta_i^t$ and a local copy of a smaller shared mentee model $S$ with a parameter set $\Theta^s$. In addition, a central server coordinates these clients for collaborative model learning. The goal is to learn a strong model in a privacy-preserving way with less communication cost.

**Federated knowledge distillation.** Next, we introduce the details of our federated knowledge distillation framework (Fig. 5). In each iteration, each client simultaneously computes the update of the local mentor model and the mentee model based on the supervision of the labeled local data, meanwhile distilling knowledge from each other in a reciprocal way with an adaptive mutual distillation mechanism. Concretely, the mentor models are locally updated, while the mentee model is shared among different clients and are learned collaboratively. Since the local mentor models have more sophisticated architectures than the mentee model, the useful knowledge encoded by the mentor model can help teach the mentee model. In addition, since the mentor model can only learn from local data while the mentee model can see the data on all clients, the mentor can also benefit from the knowledge distilled from the mentee model.

In our approach, we use three loss functions to learn mentee and mentor models locally, including an adaptive mutual distillation loss to transfer knowledge from output soft labels, an adaptive hidden loss to distill knowledge from the hidden states and self-attention heatmaps, and a task loss to directly provide task-specific supervision for learning the mentor and mentee models. We denote the soft probabilities of a sample $x_i$ predicted by the local mentor and mentee on the $i$th client as $\mathbf{y}_i^t$ and $\mathbf{y}_i^s$, respectively. Since incorrect predictions from the mentor/mentee model may mislead the other one in the knowledge transfer, we propose an adaptive method to weight the distillation loss according to the quality of predicted soft labels. We first use the task labels to compute the task losses for the mentor and mentee models (denoted as $\mathcal{L}_{t,i}^t$ and $\mathcal{L}_{s,i}^t$). We denote the gold label of $x_i$ as $\mathbf{y}_i$, and the task losses are formulated as follows:

$$\mathcal{L}_{t,i}^t = \mathrm{CE}(\mathbf{y}_i, \mathbf{y}_i^t), \tag{1}$$

$$\mathcal{L}_{s,i}^t = \mathrm{CE}(\mathbf{y}_i, \mathbf{y}_i^s), \tag{2}$$

where the binary function $\mathrm{CE}(\mathbf{a}, \mathbf{b}) = -\sum_i \mathbf{a}_i \log(\mathbf{b}_i)$ stands for cross-entropy. The adaptive distillation losses for both mentor and mentee models (denoted as $\mathcal{L}_{t,i}^d$ and $\mathcal{L}_{s,i}^d$) are formulated as follows:

$$\mathcal{L}_{t,i}^d = \frac{\mathrm{KL}(\mathbf{y}_i^s, \mathbf{y}_i^t)}{\mathcal{L}_{t,i}^t + \mathcal{L}_{s,i}^t}, \tag{3}$$

$$\mathcal{L}_{s,i}^d = \frac{\mathrm{KL}(\mathbf{y}_i^t, \mathbf{y}_i^s)}{\mathcal{L}_{t,i}^t + \mathcal{L}_{s,i}^t}, \tag{4}$$

where KL means the Kullback–Leibler divergence, i.e., $\mathrm{KL}(\mathbf{a}, \mathbf{b}) = -\sum_i \mathbf{a}_i \log(\mathbf{b}_i / \mathbf{a}_i)$. In this way, the distillation intensity is weak if the predictions of mentor and mentee are not reliable, i.e., their task losses are large. The distillation loss becomes dominant if the mentee and mentor are well tuned (small task losses), which has the potential to mitigate the risk of overfitting. In addition, previous works have validated that transferring knowledge between the hidden states[37] and hidden attention matrices[38] (if available) is beneficial for mentee teaching. Thus, taking language model distillation as an example, we also introduce additional adaptive hidden losses to align the hidden states and attention heatmaps of the mentee and the local mentors. The losses for the mentor and mentee models (denoted as $\mathcal{L}_{t,i}^h$ and $\mathcal{L}_{s,i}^h$) are formulated as follows:

$$\mathcal{L}_{t,i}^h = \mathcal{L}_{s,i}^h = \frac{\mathrm{MSE}(\mathbf{H}_i^t, \mathbf{W}_i^h \mathbf{H}^s) + \mathrm{MSE}(\mathbf{A}_i^t, \mathbf{A}^s)}{\mathcal{L}_{t,i}^t + \mathcal{L}_{s,i}^t}, \tag{5}$$

where MSE stands for the mean squared error, $\mathbf{H}_i^t$, $\mathbf{A}_i^t$, $\mathbf{H}^s$, and $\mathbf{A}^s$ respectively denote the hidden states and attention heatmaps in the $i$th local mentor and the mentee, and $\mathbf{W}_i^h$ is a learnable linear transformation matrix. Here we propose to control the intensity of the adaptive hidden loss based on the prediction correctness of the mentee and mentor. Besides, motivated by the task-specific distillation framework in[44], we also learn the mentee model based on the task-specific labels on each client. Thus, on each client the unified loss functions for computing the local update of mentor and mentee models (denoted as $\mathcal{L}_{t,i}$ and $\mathcal{L}_{s,i}$) are formulated as follows:

$$\mathcal{L}_{t,i} = \mathcal{L}_{t,i}^d + \mathcal{L}_{t,i}^h + \mathcal{L}_{t,i}^t, \tag{6}$$

$$\mathcal{L}_{s,i} = \mathcal{L}_{s,i}^d + \mathcal{L}_{s,i}^h + \mathcal{L}_{s,i}^t, \tag{7}$$

The mentee model gradients $\mathbf{g}_i$ on the $i$th client can be derived from $\mathcal{L}_{s,i}$ via $\mathbf{g}_i = \frac{\partial \mathcal{L}_{s,i}}{\partial \Theta^s}$, where $\Theta^s$ is the parameter set of mentee model. The local mentor model on each client is immediately updated by their local gradients derived from the loss function $\mathcal{L}_{t,i}$.

Afterwards, the local gradients $\mathbf{g}_i$ on each client will be uploaded to the central server for global aggregation. Since the raw model gradients may still contain some private information[45], we encrypt the local gradients before uploading. The server receives the local mentee model gradients from different clients and uses a gradient aggregator (we use the FedAvg method) to synthesize the local gradients into a global one (denoted as $\mathbf{g}$). The server further delivers the aggregated global gradients to each client for a local update. The client decrypts the global gradients to update its local copy of the mentee model. This process will be repeated until both the mentee model and the mentor model converge. Note that in the test phase, the mentor model is used for label inference.

**Algorithm 1.** FedKD

---

1: Setting the mentor learning rate $\eta_t$ and mentee learning rate $\eta_s$, client number $N$
2: Setting the hyperparameters $T_{\mathrm{start}}$ and $T_{\mathrm{end}}$
3: **for** each client $i$ (in parallel) **do**
4:　Initialize parameters $\Theta_i^t$, $\Theta^s$
5:　**repeat**
6:　　$\mathbf{g}_i^t, \mathbf{g}_i =$**LocalGradients**$(i)$
7:　　$\Theta_i^t \leftarrow \Theta_i^t - \eta_t \mathbf{g}_i^t$
8:　　$\mathbf{g}_i \leftarrow \mathbf{U}_i \sum_i \mathbf{V}_i$
9:　　Clients encrypt $\mathbf{U}_i, \sum_i, \mathbf{V}_i$
10:　Clients upload $\mathbf{U}_i, \sum_i, \mathbf{V}_i$ to the server
11:　Server decrypts $\mathbf{U}_i, \sum_i, \mathbf{V}_i$
12:　Server reconstructs $\mathbf{g}_i$
13:　Global gradients $\mathbf{g} \leftarrow 0$
14:　**for** each client $i$ (in parallel) **do**
15:　　$\mathbf{g} = \mathbf{g} + \mathbf{g}_i$
16:　**end for**
17:　$\mathbf{g} \leftarrow \mathbf{U} \sum \mathbf{V}$
18:　Server encrypts $\mathbf{U}, \sum, \mathbf{V}$
19:　Server distributes $\mathbf{U}, \sum, \mathbf{V}$ to user clients
20:　Clients decrypt $\mathbf{U}, \sum, \mathbf{V}$
21:　Clients reconstructs $\mathbf{g}$
22:　$\Theta^s \leftarrow \Theta^s - \eta_s \mathbf{g}/N$
23:　**until** Local models converges
24: **end for**
　　**LocalGradients**$(i)$:
25: Compute task losses $\mathcal{L}_{t,i}^t$ and $\mathcal{L}_{s,i}^t$
26: Compute losses $\mathcal{L}_{t,i}^d$, $\mathcal{L}_{s,i}^d$, $\mathcal{L}_{t,i}^h$, and $\mathcal{L}_{s,i}^h$
27: $\mathcal{L}_i^t \leftarrow \mathcal{L}_{t,i}^t + \mathcal{L}_{t,i}^d + \mathcal{L}_{t,i}^h$
28: $\mathcal{L}_i^s \leftarrow \mathcal{L}_{s,i}^t + \mathcal{L}_{s,i}^d + \mathcal{L}_{s,i}^h$
29: Compute local mentor gradients $\mathbf{g}_i^t$ from $\mathcal{L}_i^t$
30: Compute local mentee gradients $\mathbf{g}_i$ from $\mathcal{L}_i^s$
31: **return** $\mathbf{g}_i^t, \mathbf{g}_i$

---

**Dynamic gradients approximation.** In our FedKD framework, although the size of mentee model updates is smaller than the mentor models, the communication

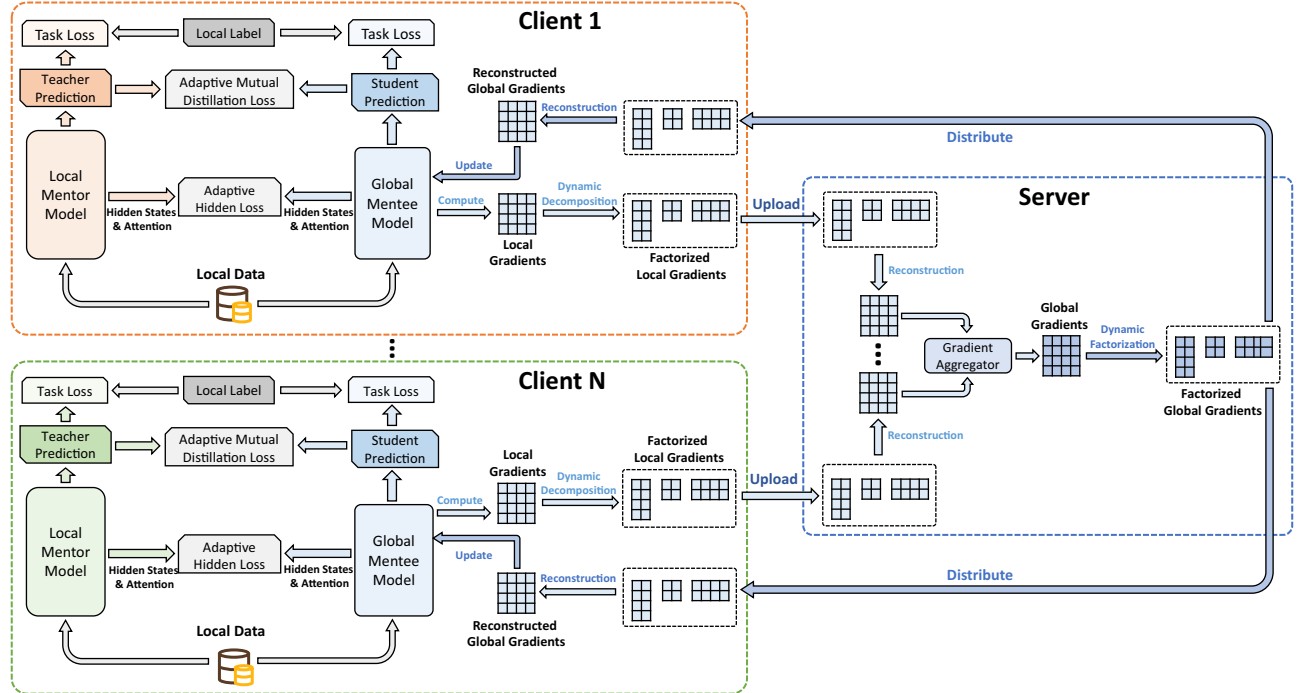

**Fig. 5 The framework of our FedKD approach.** The local data is used to train the local mentor model and global mentee model. Both models are learned from local labeled data as well as the prediction and hidden results of each other. The local gradients are decomposed before uploading to the server, and then reconstructed on the server for aggregation. The aggregated global gradients are further decomposed and distributed to clients for local updates.

cost can still be relatively high when the mentee model is not tiny. Thus, we explore to further compress the gradients exchanged among the server and clients to reduce computational cost. Motivated by the low-rank properties of model parameters[46], we use SVD to factorize the local gradients into smaller matrices before uploading them. The server reconstructs the local gradients by multiplying the factorized matrices before aggregation. The aggregated global gradients are further factorized, which are distributed to the clients for reconstruction and model update. More specifically, we denote the gradient $\mathbf{g}_i \in \mathbb{R}^{P \times Q}$ as a matrix with $P$ rows and $Q$ columns (we assume $P \geq Q$). It is approximately factorized into the multiplication of three matrix, i.e., $\mathbf{g}_i \approx \mathbf{U}_i \sum_i \mathbf{V}_i$, where $\mathbf{U}_i \in \mathbb{R}^{P \times K}$, $\sum_i \in \mathbb{R}^{K \times K}$, $\mathbf{V}_i \in \mathbb{R}^{K \times Q}$ are factorized matrices and $K$ is the number of retained singular values. If the value of $K$ satisfies $PK + K^2 + KQ < PQ$, the size of the uploaded and downloaded gradients can be reduced. Note that we formulate $\mathbf{g}_i$ as a single matrix for simplicity. In practice, different parameter matrices in the model are factorized independently, and the global gradients on the server are factorized in the same way. We denote the singular values of $\mathbf{g}_i$ as $[\sigma_1, \sigma_2, \ldots, \sigma_Q]$ (ordered by their absolute values). To control the approximation error, we use an energy threshold $T$ to decide how many singular values are kept, which is formulated as follows:

$$\min_K \frac{\sum_{i=1}^{K} \sigma_i^2}{\sum_{i=1}^{Q} \sigma_i^2} > T. \qquad (8)$$

To better help the model converge, we propose a dynamic gradient approximation strategy by using a dynamic value of $T$. The function between the threshold $T$ and the percentage of training steps $t$ is formulated as follows:

$$T(t) = T_{\text{start}} + (T_{\text{end}} - T_{\text{start}})t, t \in [0, 1], \qquad (9)$$

where $T_{\text{start}}$ and $T_{\text{end}}$ are two hyperparameters that control the start and end values of $T$. In this way, the mentee model is learned on roughly approximated gradients at the beginning, while learned on more accurately approximated gradients when the model gets to convergence, which can help learn a more accurate mentee model.

To help readers better understand how FedKD works, we summarize the entire workflow of FedKD (Algorithm 1).

**Complexity analysis**. In this section, we present some analysis on the complexity of our FedKD approach in terms of computation and communication cost. We denote the number of communication rounds as $R$ and the average data size of each client as $D$. Thus, the computational cost of directly learning a large model (the parameter set is denoted as $\Theta^t$) in a federated way is $O(RD|\Theta^t|)$, and the communication cost is $O(R|\Theta^t|)$ (we assume the cost is linearly proportional to model sizes). In FedKD, the communication cost is $O(R|\Theta^s|/\rho)$ ($\rho$ is the gradient compression ratio), which is much smaller because $|\Theta^s| \ll |\Theta^t|$ and $\rho > 1$. The computational cost contains three parts, i.e., local mentor model learning, mentee

model learning and gradient compression/reconstruction, which are $O(RD|\Theta^t|)$, $O(RD|\Theta^s|)$ and $O(RPQ^2)$, respectively. The total computational cost of FedKD is $O(RD|\Theta^t| + RD|\Theta^s| + RPQ^2)$. In practice, compared with the standard FedAvg[4] method, the extra computational cost of learning the mentee model in FedKD is much smaller than learning the large mentor model, and SVD can also be very efficiently computed in parallel. Thus, FedKD is much more communication-efficient than the standard FedAvg method and meanwhile does not introduce much computational cost.

**Reporting summary**. Further information on research design is available in the Nature Research Reporting Summary linked to this article.

## Data availability
The datasets that support the findings of this study are all available ones, and the use of them in this work adheres to the licenses of these datasets. The MIND dataset is available at https://msnews.github.io/. The ADR dataset is available at https://healthlanguageprocessing.org/smm4h18. The $C_{ADEC}$ dataset is available at https://data.csiro.au. The ADE dataset is available at https://sites.google.com/site/adecorpus/. The SMM4H dataset is available at https://healthlanguageprocessing.org/smm4h19. Source Data are provided with this paper.

## Code availability
Codes used for this study are available on a public repository https://github.com/wuch15/FedKD[47]. In addition, all experiments and implementation details are described in sufficient detail in the Methods section and in the Supplementary Information.

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

## Acknowledgements

We thank Tao Qi and Ruixuan Liu for their discussions on this work. This work was supported by the National Natural Science Foundation of China under Grant numbers 2021ZD0113902 (Y.H.), U1936208 (Y.H.), U1836204 (Y.H.), U1936216 (Y.H.), and the research initiation project of Zhejiang Lab under Grant number 2020LC0PI01 (C.W.).

## Author contributions

Y.H. coordinated the research project and supervised the project with assistance from X.X. C.W. implemented the models in the FedKD framework and conducted experiments. C.W., F.W. and L.L. discussed and analyzed the results. C.W., F.W. and L.L. contributed to the writing of the manuscript with assistance from Y.H. and X.X. All authors contributed to the discussion and design of the FedKD framework.

## Competing interests

The authors declare the following competing interests. F.W. and X.X. currently are employees at Microsoft Research Asia and hold the positions of researcher. L.L. is currently an employee at Sony AI and holds the position of researcher. No author holds substantial shares in these companies. The authors declare no other types of competing interests.
