## [Peer Review File · Nature Communications]

Reviewers' Comments:

Reviewer #1:

Remarks to the Author:

Paper Summary:

This paper proposes to use knowledge distillation for both personalization and communication reduction. Each client has a local student model and a teacher model where the student model is shared and the teacher models are personalized. This work further reduces communication by gradient compression through factorization. The experiments validate the effectiveness of the proposed approaches.

Comments:

- * I like the problem of thinking about model compression and personalization simultaneously. Although this work does not explicitly mention it, it is training different but related models for each client (i.g., performing personalization).
- * All components of the proposed method are known and well-studied in previous works (either in or out of the context of federated learning). The gradient compression approach is standard. Hence, it is difficult (for me) to learn any new insight from this work.
- * The major flaw of this work, in my mind, is the cost of local training of large teacher models. I doubt if this method would be useful in practice for memory-constrained devices as training large models is not feasible. Related to this, I think the scope of this study needs to be calibrated--- discussing which setting of federated learning this work focuses on: cross-silo or cross-device. In cross-device settings (with potentially millions of edge devices), people care about communication, and the devices are often unreliable (so the algorithm needs to adapt to this setting with partial device participation, which is not supported in Algorithm 1). In cross-silo settings (learning across hospitals or banks), it is more realistic to assume large models can be trained locally. The targeted setting also affects of choices of datasets for empirical evaluation.
- * I suggest bringing personalization into the whole story. Currently, it is buried under the methods and experiments, but this is a critical aspect of this work.
- * Table 1: Compared with Fed, the only benefit of FedKD is compression, but I expect to see the benefits of personalization. Is it because the data are uniformly randomly partitioned (which means each client has the training data from the same distribution) so learning a single global model is good (without the need to adapt to each client's own distribution)? It is not clear how the data are partitioned across clients.
- * In addition, for the application of news recommendation, there should be thousands (or more) clients/users in practice, but there are only a few in the current experiments. In general, I feel the datasets or experiments should be set up in a better way to model the real-world scenarios (see e.g., <https://arxiv.org/pdf/2107.06917.pdf> [first few sections]).
- * Related to the personalization comment above, the claim: 'Different from them, the performance of FedKD is even comparable with learning a big model on centralized data' is not fair, as in FedKD, each client has its own model; but for the baselines, there is only one single model to fit all data.

Reviewer #2:

Remarks to the Author:

Overview:

This work focus on using knowledge distillation (KD) and matrix approximation for federated

learning. I find this paper particularly interesting as KD fits this scenario perfectly: the teacher model can only access the local copy of data and the global student model can access all updates, thus they could learn from each other and improve their prediction accuracy. In addition, the code is also provided with the paper, and results could be easily reproduced.

Overall this paper proposes a very promising direction that could save a significant amount of energy, and help "Green AI" possible.

Strength:

1. As I mentioned in the overview part, the idea of the paper is novel and interesting. In addition, it fits the scenario pretty well as the student and teacher models could learn and improve from each other with the proposed MD loss.

2. The results presented in table 1 are surprisingly good. The proposed method could achieve similar results to the upper bound (UniLM (Cen)) with only 5~10% of communication cost. This framework does not rely on the base method (UniLM in this case) and potentially could benefit many other applications in the real world.

Questions/Concerns:

1. I am very familiar with KD and all KD baseline methods in table 1, but not federated learning. It still seems there are many baselines in KD yet several baselines in federated learning are missing, such as SCAFFOLD and etc.

2. Still in table 1, if you could add an extra number in communication cost stating the actual deduction compare to baseline, it is easier for the reader to understand. For example, "1.03GB" can be written as "1.03Gb(2x)". In addition, some results could be bold or underscored if they are the best or second-best in the respective columns (excluding the upper bound, maybe)

3. The teacher model's performance dropped significantly compare to student model in figure 2, and the explanation in the paper is "help student better imitate the complicated teacher models, and can help teachers break the limitation of the amount of local labeled data". Maybe add more detailed results to support this claim.

4. The paper basically proposed two orthogonal approaches that are based on KD and SVD, respectively. However, no detailed ablation study (including AUC, score and communication cost) is presented, e.d. FedKD vs FedKD - KD vs FedKD - SVD. In figure 2, FedKD - KD is discussed with respect to the accuracy-related metric. However, how much does it contribute to the communication cost?

5. In line 196 it mentions "and the knowledge distilled from each other", which I find confusing because MD has not been introduced yet.

6. In line 211, maybe add more intuition that why g_i is in low-rank and therefore can be approximated using SVD. In line 123, it briefly touches the intuition, but maybe adds more detailed results to support this claim.

7. Figure 5 is not very clear to general reader as usage of color and shape are confusion.

Reviewer #3:

Remarks to the Author:

The paper proposes a novel federated learning method called FedKD for reducing communication costs during training. FedKD includes two main techniques. Firstly, each client learns two models, a large one and a smaller one. The larger is trained on local data while the smaller one is used to coordinate learning between clients. The two networks distill knowledge to each other. Moreover, the gradient exchanged between clients and server is compressed using SVD. Both distillation and

SVD-based compression are done in a dynamic way depending respectively on model performance and training step.

The proposed method seems to be novel as far as I know, although I am not completely up to date with federated learning methods. The proposed techniques are interesting and seem to make sense for reducing communication costs. The overall writing is generally clear, although it could be improved a little bit (see below). I think that some information about the experimental evaluation is missing and I believe that the comparisons with relevant federated learning baselines for reducing communication costs could be improved.

The good experimental performance of FedKD seems to mainly come from the strong base model used in the paper: UniLM. Interestingly, the authors manage to maintain this good performance while significantly reducing the communication cost. Since the overall proposed method seems to be generic, it would have been interesting to evaluate it with other base models to verify that it does not depend on UniLM, i.e., it can also preserve the performance of other models while reducing communication costs.

I find that some details are missing to fully appreciate the experimental results:

- The authors specify that they use the teacher model for evaluation because it obtains better performance. Since one teacher model is learned per client, which one is used for the evaluation? Or do you average the results over all the models?
- In Table 1, are DistilBERT, BERT-PKD, Tiny-BERT trained using FedAvg?
- In Table 1 and Fig. 1, is the same base model (i.e., UniLM) used in FetchSGD and FedDropout?
- Why did the authors choose FetchSGD and FedDropout as baselines? And not, for instance, Feded or Fedmd? Although I'm not up to date with the research work on reducing communication costs in federated learning, it seems that it is a very active research direction. I think more discussion about the choice of the baselines and/or more experiments with other baselines would have been appreciated.

Since the contribution of the paper is algorithmic, I believe that the authors should have shared the code with the reviewers. Also, I would suggest them to share their code on a public repository to ease the reproducibility of their work.

A minor remark, since there is mutual distillation, the student model also teaches the teacher model, which does not match well the current terminology in my opinion. Instead of teacher and student models, I would suggest the authors to find better names for the larger and the smaller models.

Detailed comments:

Citations should not be part of sentences, especially when citations are superscripts. Also, you may want to differentiate citations and footnotes.

Lines 31-32: are there currently models with trillions of parameters?

Lines 47-48: please recall the Protégé effect

Line 72: rephrase the sentence. Currently, it sounds grammatically incorrect.

Line 78: please reorganize your list of baselines. It is currently hard to parse and the list is interrupted by whole sentences

Line 83: the the

Fig. 4: Is it a newly-observed phenomenon? If not, please give proper citations.

The last paragraph of page 2 and the first paragraph of the Discussion section are very similar. I think it would be best to merge those paragraphs and use the gained space to provide some information that is currently missing about the experiments.

Line -2 above (1): the superscript of the second loss is incorrect

Line 229: I think the best practice for code sharing is to make them available on a public repository.

Line 232: the use of them -> their use

We sincerely thank the editor and all anonymous reviewers for their great effort in reviewing this paper and providing insightful and constructive comments. We have carefully studied the comments and made corresponding revisions to our paper. Our detailed responses to the reviewers' comments are listed as follows.

General Notes:

According to the suggestion of Reviewer 3, we change the terms “teacher” and “student” to “mentor” (large model) and “mentee” (small model). This is because in our mutual distillation method the small model does not only passively receive knowledge from the teacher. We use “mentor” and “mentee” to describe the models' behaviors in our approach, which indicate that they share knowledge reciprocally while the “mentor” is more senior than the “mentee” since the “mentor” has a more complicated model architecture.

Responses to Reviewer 1:

We sincerely thank the reviewer for providing the insightful comments and constructive suggestions. Our responses are as follows.

- **Comment 1:** I like the problem of thinking about model compression and personalization simultaneously. Although this work does not explicitly mention it, it is training different but related models for each client (i.g., performing personalization). I suggest bringing personalization into the whole story. Currently, it is buried under the methods and experiments, but this is a critical aspect of this work.
- **Response 1:** This is a very insightful suggestion. We now explicitly mention the personalization aspect of our work in the story and discuss its potential in the discussion part. Thanks a lot for this comment.

- **Comment 2:** All components of the proposed method are known and well-studied in previous works (either in or out of the context of federated learning). The gradient compression approach is standard. Hence, it is difficult (for me) to learn any new insight from this work.
- **Response 2:** Thanks for the comment. Although the concepts of knowledge distillation and gradient compression are not newly proposed, to our best knowledge the adaptive mutual distillation and dynamic gradient approximation methods are not studied by prior works. In addition, we have several new findings from our experiments. Thus, our work can provide useful insight for researchers to develop communication-efficient federated learning systems, better transfer useful knowledge among models, and have a deeper understanding of the evolutionary low-rank property of deep model update.

- **Comment 3:** The major flaw of this work, in my mind, is the cost of local training of large teacher models. I doubt if this method would be useful in practice for memory-constrained devices as training large models is not feasible.
- **Response 3:** We totally agree with the reviewer that locally training large models may bring high overheads for edge devices. However, how to efficiently train large models on low-resource

devices via federated learning is still an open problem. Though there are some recent attempts such as [1], they still cannot handle very large models. Thus, We plan to address this problem in our future work.

[1] Heterofl: Computation and communication efficient federated learning for heterogeneous clients. arXiv preprint arXiv:2010.01264.

- **Comment 4:** I think the scope of this study needs to be calibrated—discussing which setting of federated learning this work focuses on: cross-silo or cross-device. In cross-device settings (with potentially millions of edge devices), people care about communication, and the devices are often unreliable (so the algorithm needs to adapt to this setting with partial device participation, which is not supported in Algorithm 1). In cross-silo settings (learning across hospitals or banks), it is more realistic to assume large models can be trained locally. The targeted setting also affects of choices of datasets for empirical evaluation.
- **Response 4:** Very good suggestion. We agree that our current setting is more feasible in cross-silo scenarios, which is consistent with the choices of datasets and experimental settings. We have clarified the focus of this work in the last paragraph of the first section. Nevertheless, if the involved devices are strong enough for learning local models (e.g., relatively stronger PCs), our approach can still be applied to cross-device settings. We have added more discussions on the application scope of our work in the Discussion section to clarify this point.

- **Comment 5:** Table 1: Compared with Fed, the only benefit of FedKD is compression, but I expect to see the benefits of personalization. Is it because the data are uniformly randomly partitioned (which means each client has the training data from the same distribution) so learning a single global model is good (without the need to adapt to each client's own distribution)? It is not clear how the data are partitioned across clients.
- **Response 5:** In Table 1, the data is randomly partitioned across clients, which means that the data is iid. By contrast, we also compare the results of different methods under non-iid settings (Figure 1), where different clients keep different genres of data (the first paragraph in the Performance Evaluation section). We can observe a notable performance gain over other non-personalized approach, which emphasizes the importance of personalized parameters in handling heterogeneous data.

- **Comment 6:** In addition, for the application of news recommendation, there should be thousands (or more) clients/users in practice, but there are only a few in the current experiments. In general, I feel the datasets or experiments should be set up in a better way to model the real-world scenarios (see e.g., <https://arxiv.org/pdf/2107.06917.pdf> [first few sections]).
- **Response 6:** Thanks for the comment. As described in Response 4, our experimental setting more fits the cross-silo scenarios, and therefore only a few clients are used. We plan to study how to generalize our approach to learning models effectively and efficiently on a large number of devices whose resources and local data are quite limited.

- **Comment 7:** Related to the personalization comment above, the claim: 'Different from them, the performance of FedKD is even comparable with learning a big model on centralized data'

is not fair, as in FedKD, each client has its own model; but for the baselines, there is only one single model to fit all data.

- **Response 7:** Thanks for the comment. We agree that the improvement of FedKD is partially due to the personalized parameters. Nevertheless, it can indeed achieve better performance than other baseline methods. This in fact shows the potentials of personalized parameters in retaining the performance of communication-efficient federated learning. We have added discussions on this point to our paper.

Responses to Reviewer 2:

We sincerely thank the reviewer for providing the insightful comments and constructive suggestions. Our responses are as follows.

- **Comment 1:** I am very familiar with KD and all KD baseline methods in table 1, but not federated learning. It still seems there are many baselines in KD yet several baselines in federated learning are missing, such as SCAFFOLD and etc.
- **Response 1:** Thank you for the great comment. We added an additional state-of-the-art KD-based compression method MiniLM for comparison. In addition, we added two federated learning baselines, including SCAFFOLD [2] and FedPAQ [3]. We choose these baselines because they use two other ways for reducing communication costs, i.e., accelerating the model convergence to reduce the communication rounds, and quantizing the communicated parameters to reduce the cost. We find although SCAFFOLD can reduce the rounds of communication on the MIND dataset, it even increases the communication cost because it needs to maintain and exchange additional control variate. In addition, we find FedPAQ can reduce the communication cost to some extent, but its performance sacrifice is large when the number of float bits is insufficient. FedKD can achieve a better tradeoff between performance and communication cost than other baseline methods.

[2] Scaffold: Stochastic controlled averaging for federated learning. ICML 2020.

[3] Fedpaq: A communication-efficient federated learning method with periodic averaging and quantization. AISTATS 2020.

- **Comment 2:** Still in table 1, if you could add an extra number in communication cost stating the actual deduction compare to baseline, it is easier for the reader to understand. For example, "1.03GB" can be written as "1.03Gb(2x)". In addition, some results could be bold or underscored if they are the best or second-best in the respective columns (excluding the upper bound, maybe)
- **Response 2:** Thank you for your great suggestion on improving the presentation. We have revised Table 1 according to your kind suggestion.
- **Comment 3:** The teacher model's performance dropped significantly compare to student model in figure 2, and the explanation in the paper is "help student better imitate the complicated teacher models, and can help teachers break the limitation of the amount of local labeled data". Maybe add more detailed results to support this claim.

- **Response 3:** Thanks for the comment. The performance of teacher model is lower than the student model because the teacher model can only learn from the local data while the student model is learned on decentralized data. Nevertheless, the teacher model has more complicated architectures and can learn more sophisticated features. Thus, knowledge distillation can help the student model learn useful knowledge from the teacher. In addition, since the student model can encode complementary knowledge from different clients, the teacher model can also learn from the student model. From the results in Supplementary Figure 1, we can see that when the number of clients increases from 1 to 3, the performance can even slightly increase, even the number of local data becomes smaller. This is because different clients keep different teacher models, and thereby can encode knowledge in different views. The student model can serve as a carrier of knowledge on multiple clients to further enhance the local teacher models. We have refined the discussions on this phenomenon to improve the clarity.

- **Comment 4:** The paper basically proposed two orthogonal approaches that are based on KD and SVD, respectively. However, no detailed ablation study (including AUC, score and communication cost) is presented, e.d. FedKD vs FedKD - KD vs FedKD - SVD. In figure 2, FedKD - KD is discussed with respect to the accuracy-related metric. However, how much does it contribute to the communication cost?
- **Response 4:** Thanks for the comment. The communication cost and AUC score of FedKD without SVD are shown in Supplementary Figure 2(a) (when $T_{start}=1$). According to your suggestion, we have added an additional ablation study to show the influence of KD and SVD on the communication cost and AUC scores in Supplementary Figure 3 (different from the results in Figure 2 with only mutual KD removed, this figure shows the results without KD). The results indicate that KD has a major contribution to improving the performance, while SVD compression has a major contribution to the decrease of communication costs. Thus, combining them can keep good model performance and reduce the communication cost.

- **Comment 5:** In line 196 it mentions "and the knowledge distilled from each other", which I find confusing because MD has not been introduced yet.
- **Response 5:** Thanks for pointing out this issue. We have revised this sentence to briefly introduce the idea of MD first.

- **Comment 6:** In line 211, maybe add more intuition that why g_i is in low-rank and therefore can be approximated using SVD. In line 123, it briefly touches the intuition, but maybe adds more detailed results to support this claim.
- **Response 6:** Thanks for the insightful comment. In recent years researchers find the over-parameterization property of deep neural networks leads to the low-rank characteristics of their parameters and model updates [4]. Motivated by their observations, we think the model gradients can be approximated via SVD. We have added some references to show the intuition behind our approach.

[4] On learning over-parameterized neural networks: A functional approximation perspective. NeurIPS 2019.

- **Comment 7:** Figure 5 is not very clear to general reader as usage of color and shape are confusion.
- **Response 7:** Thanks a lot for the comment. We have refined the use of element shapes and colors to make them easier to understand. More specifically, we use the same shapes for different losses and for different model predictions/labels. We change all colors of gradients to blue for better consistency and use different colors for different local mentor (teacher) models.

Responses to Reviewer 3:

We sincerely thank the reviewer for providing the insightful comments and constructive suggestions. Our responses are as follows.

- **Comment 1:** The proposed method seems to be novel as far as I know, although I am not completely up to date with federated learning methods. The proposed techniques are interesting and seem to make sense for reducing communication costs. The overall writing is generally clear, although it could be improved a little bit (see below). I think that some information about the experimental evaluation is missing and I believe that the comparisons with relevant federated learning baselines for reducing communication costs could be improved.
- **Response 1:** Thank you very much for the kind words and constructive suggestions. We have supplied the missing details and added more baselines to compare. We hope our revisions can further improve the clarity of our paper.
- **Comment 2:** The good experimental performance of FedKD seems to mainly come from the strong base model used in the paper: UniLM. Interestingly, the authors manage to maintain this good performance while significantly reducing the communication cost. Since the overall proposed method seems to be generic, it would have been interesting to evaluate it with other base models to verify that it does not depend on UniLM, i.e., it can also preserve the performance of other models while reducing communication costs.
- **Response 2:** This is a great suggestion. We added a comparison of different base models (Supplementary Figure 4). The results show that FedKD can consistently retain the performance of different models meanwhile reduce the communication costs.
- **Comment 3:** The authors specify that they use the teacher model for evaluation because it obtains better performance. Since one teacher model is learned per client, which one is used for the evaluation? Or do you average the results over all the models?
- **Response 3:** Good point. Each client uses its own teacher model for evaluation, and we average the results of the teacher models on different clients. We added these details to the evaluation part.
- **Comment 4:** In Table 1, are DistilBERT, BERT-PKD, Tiny-BERT trained using FedAvg?
- **Response 4:** Yes. They are finetuned with FedAvg. We added this detail to our paper.
- **Comment 5:** In Table 1 and Fig. 1, is the same base model (i.e., UniLM) used in FetchSGD and FedDropout?

- **Response 5:** Yes. We use the same base model for fair comparison. We have added some explanations to our paper to clarify this point.

- **Comment 6:** Why did the authors choose FetchSGD and FedDropout as baselines? And not, for instance, Feded or Fedmd? Although I'm not up to date with the research work on reducing communication costs in federated learning, it seems that it is a very active research direction. I think more discussion about the choice of the baselines and/or more experiments with other baselines would have been appreciated.
- **Response 6:** Good point. We choose to compare FetchSGD and FedDropout because they use two representative ways to reduce communication cost, i.e., using sketched model updates and reducing communicated model parameters. We do not compare Feded and Fedmd because they rely on additional public unlabeled data, which is often unavailable due to privacy reasons (e.g., even unlabeled user data is highly privacy-sensitive). Following your suggestion, we compare two additional baselines, i.e., SCAFFOLD and FedPAQ. They use other two ways including convergence acceleration and quantization. We have also added the discussions on the selection of baselines to the "Performance Evaluation" Section.

- **Comment 7:** Since the contribution of the paper is algorithmic, I believe that the authors should have shared the code with the reviewers. Also, I would suggest them to share their code on a public repository to ease the reproducibility of their work.
- **Response 7:** Thanks for the suggestion. We have made our code publicly available on a GitHub repository (<https://github.com/wuch15/FedKD>).

- **Comment 8:** A minor remark, since there is mutual distillation, the student model also teaches the teacher model, which does not match well the current terminology in my opinion. Instead of teacher and student models, I would suggest the authors to find better names for the larger and the smaller models.
- **Response 8:** This is a great suggestion. We totally agree with you about the terminology of large and small models in mutual distillation. We change their names to "mentor" (large model) and "mentee" (small model), which indicate that they share knowledge reciprocally and the "mentor" is more senior than the "mentee".

- **Comment 9:**
Citations should not be part of sentences, especially when citations are superscripts. Also, you may want to differentiate citations and footnotes.
- **Response 9:** Thanks a lot for the comment. We have carefully refined the use of citations in our paper and avoided using them as part of sentences. We also place the footnote in the paragraphs since they have similar formats with citations.

- **Comment 10:**
Lines 31-32: are there currently models with trillions of parameters?
- **Response 10:** Yes. To our best knowledge, the currently largest model has up to 100 trillion parameters [5]. We have added references to support this claim.

[5] Persia: A Hybrid System Scaling Deep Learning Based Recommenders up to 100 Trillion Parameters. arXiv preprint arXiv:2111.05897.

- **Comment 11:** Lines 47-48: please recall the Protégé effect
- **Response 11:** Thanks for the suggestion. We have added a quote "*while we teach, we learn*" attributed to Seneca the Younger to explain this effect.

- **Comment 12:** Line 78: please reorganize your list of baselines. It is currently hard to parse and the list is interrupted by whole sentences
- **Response 12:** Thanks a lot for the suggestion. We have reorganized the baseline list and added numbers to indicate their orders.

- **Comment 13:** Fig. 4: Is it a newly-observed phenomenon? If not, please give proper citations.
- **Response 13:** We agree that the low-rank property of neural models is widely studied, and we added some references to support the findings in Fig. 4. However, to our best knowledge there is no prior work showing the dynamic rank change and the differences between different types of parameter matrices. Thus, we think Fig. 4 shows some newly-observed phenomena.

- **Comment 14:** The last paragraph of page 2 and the first paragraph of the Discussion section are very similar. I think it would be best to merge those paragraphs and use the gained space to provide some information that is currently missing about the experiments.
- **Response 14:** Great suggestion. We have simplified the first paragraph of the Discussion section and enriched the details of experiments according to your suggestions.

- **Comment 15:** Some typos and grammatical issues.
- **Response 15:** Thank you very much for pointing out these issues. We have corrected them and carefully refined the writing of this article.

Reviewers' Comments:

Reviewer #1:

Remarks to the Author:

Thanks to the authors for their responses to my previous reviews. However, some of my concerns still remain:

* Regarding novelty of this work: Gradient factorization is a popular technique for communication-efficient training (e.g., Pufferfish: Communication-efficient Models At No Extra Cost, Initialization and Regularization of Factorized Neural Layers)

The 'approximate' gradient factorization in this paper adds a linear scaling of the number of preserved singular values to better control the training dynamics. However, I still think the novelty of such an addition is limited. Similarly, the mutual distillation idea has appeared in previous works, and the 'adaptive' aspect proposed in this work is a bit ad-hoc and not principled enough.

* Regarding settings and assumptions: If this work focuses on cross-silo settings, then communication is usually not a critical problem, which limits the impact of proposed gradient compression methods.

Reviewer #2:

Remarks to the Author:

My original reviews are posted below. I am very pleased with the authors' responses and revisions and think it will be beneficial to the community if the paper could get published.

Overview:

This work focus on using knowledge distillation (KD) and matrix approximation for federated learning. I find this paper particularly interesting as KD fits this scenario perfectly: the teacher model can only access the local copy of data and the global student model can access all updates, thus they could learn from each other and improve their prediction accuracy. In addition, the code is also provided with the paper, and results could be easily reproduced.

Overall this paper proposes a very promising direction that could save a significant amount of energy, and help "Green AI" possible.

Strength:

1. As I mentioned in the overview part, the idea of the paper is novel and interesting. In addition, it fits the scenario pretty well as the student and teacher models could learn and improve from each other with the proposed MD loss.

2. The results presented in table 1 are surprisingly good. The proposed method could achieve similar results to the upper bound (UniLM (Cen)) with only 5~10% of communication cost. This framework does not rely on the base method (UniLM in this case) and potentially could benefit many other applications in the real world.

Questions/Concerns:

1. I am very familiar with KD and all KD baseline methods in table 1, but not federated learning. It still seems there are many baselines in KD yet several baselines in federated learning are missing, such as SCAFFOLD and etc.

2. Still in table 1, if you could add an extra number in communication cost stating the actual deduction compare to baseline, it is easier for the reader to understand. For example, "1.03GB" can be written as "1.03Gb(2x)". In addition, some results could be bold or underscored if they are the best or second-best in the respective columns (excluding the upper bound, maybe)

3. The teacher model's performance dropped significantly compare to student model in figure 2, and the explanation in the paper is "help student better imitate the complicated teacher models, and can help teachers break the limitation of the amount of local labeled data". Maybe add more detailed results to support this claim.

4. The paper basically proposed two orthogonal approaches that are based on KD and SVD, respectively. However, no detailed ablation study (including AUC, score and communication cost) is presented, e.d. FedKD vs FedKD - KD vs FedKD - SVD. In figure 2, FedKD - KD is discussed with respect to the accuracy-related metric. However, how much does it contribute to the communication cost?

5. In line 196 it mentions "and the knowledge distilled from each other", which I find confusing because MD has not been introduced yet.

6. In line 211, maybe add more intuition that why g_i is in low-rank and therefore can be approximated using SVD. In line 123, it briefly touches the intuition, but maybe adds more detailed results to support this claim.

7. Figure 5 is not very clear to general reader as usage of color and shape are confusion.

Reviewer #3:

Remarks to the Author:

I would like to thank the authors for addressing my previous comments. I think the update of the manuscript makes the content clearer now, notably in terms of the contributions. Moreover, the new experiments demonstrate that the authors' general framework indeed contributes to maintaining a good performance, while reducing the communication costs. Overall, I feel my previous concerns are addressed.

Small typos:

Line 85: The second group is finetuning -> The second group finetunes

Lines 137-138: are in low-rank -> are low-rank

Line 154: decentralized

We sincerely thank the editor and reviewers for their great effort in reviewing this paper and providing insightful and constructive comments. We have carefully studied the comments and revised our paper. Our detailed responses to the reviewers' comments are listed as follows.

Responses to Reviewer 1:

We sincerely thank the reviewer for reviewing our article again and providing constructive feedback. Our responses to your remaining concerns are as follows.

- **Comment 1:** Regarding novelty of this work: Gradient factorization is a popular technique for communication-efficient training (e.g., Pufferfish: Communication-efficient Models At No Extra Cost, Initialization and Regularization of Factorized Neural Layers). The “approximate” gradient factorization in this paper adds a linear scaling of the number of preserved singular values to better control the training dynamics. However, I still think the novelty of such an addition is limited. Similarly, the mutual distillation idea has appeared in previous works, and the “adaptive” aspect proposed in this work is a bit ad-hoc and not principled enough.
- **Response 1:** Thanks for the comment. We agree that gradient factorization is a well-explored technique (also discussed in our article). However, in this work we find the phenomenon of the evolution of gradient ranks, which motivates our proposed method. We believe this finding can also inspire researchers in other fields to better understand deep model learning. In addition, the adaptive mechanism of our mutual distillation method is not limited to federated learning, and can be generalized to various machine learning scenarios that involve knowledge transfer among multiple models. Thus, it has the potential to empower many other tasks without heavy effort.
- **Comment 2:** Regarding settings and assumptions: If this work focuses on cross-silo settings, then communication is usually not a critical problem, which limits the impact of proposed gradient compression methods.
- **Response 2:** Thanks for the comment. Although this work is targeted at cross-silo settings, the communication among platforms can still be a bottleneck for federated learning because many recent deep models are extremely large. In addition, our approach can also be generalized to cross-device settings when local devices are capable of locally maintaining models with moderate sizes. Thus, we think the proposed methods have the potential to empower both cross-silo and cross-device federated learning scenarios.

Responses to Reviewer 2:

We sincerely thank the reviewer for reviewing this paper again and providing positive feedback.

Responses to Reviewer 3:

We sincerely thank the reviewer for the kind words and suggestions. Our responses are as follows.

- **Comment 1:** Small typos: Line 85: The second group is finetuning -> The second group finetunes, Lines 137-138: are in low-rank -> are low-rank, Line 154: decentralized
- **Response 1:** Thank you very much for pointing out these small flaws. We have corrected them and refined the paper carefully according to journal guidelines.